

# An affordable and automated imaging approach to acquire highly resolved individual data—an example of copepod growth in response to multiple stressors

Jan Heuschele, Torben Lode, Tom Andersen, Katrine Borgå and
Josefin Titelman

Department of Biosciences, University of Oslo, Oslo, Norway

## ABSTRACT

Individual trait variation is essential for populations to cope with multiple stressors and continuously changing environments. The immense number of possible stressor combinations and the influence of phenotypic variation makes experimental testing for effects on organisms challenging. The acquisition of such data requires many replicates and is notoriously laborious. It is further complicated when responses occur over short time periods. To overcome such challenges, we developed an automated imaging platform to acquire temporally highly resolved individual data. We tested this platform by exposing copepods to a combination of a biotic stressor (predator cues) and a toxicant (copper) and measured the growth response of individual copepods. We tested the automatically acquired data against published manually acquired data with much lower temporal resolution. We find the same general potentiating effects of predator cues on the adverse effects of copper, and the influence of an individual's clutch identity on its ability to resist stress, between the data obtained from low and high temporal resolution. However, when using the high temporal resolution, we also uncovered effects of clutch ID on the timing and duration of stage transitions, which highlights the importance of considering phenotypic variation in ecotoxicological testing. Phenotypic variation is usually not acknowledged in ecotoxicological testing. Our approach is scalable, affordable, and adjustable to accommodate both aquatic and terrestrial organisms, and a wide range of visually detectable endpoints. We discuss future extensions that would further widen its applicability.

# INTRODUCTION

The life of any organism is a continuous struggle with different stressors, be it from other organisms or the physical environment. Since the last century organisms are also exposed to novel artificial substances of anthropogenic origin, such as chemical toxicants, as well as rapid changes in the environment due to human activities. In nature, stressors never act independently of one another. In addition to anthropogenic stressors, organisms

Corresponding author
Jan Heuschele, janheu@ibv.uio.no

must cope with natural biotic stress from parasites, competition for resources, and predation risk. While direct consumption is detrimental for prey, non-consumptive effects are complex (*Heuschele et al., 2014*), and include behavioural, morphological, and physiological changes (*Verity & Smetacek, 1996*). Most studies on mixture toxicity and multiple stressors focus either on interactive effects of two toxicants, or one pollutant in interaction with changes in the physical environment (*Gunderson, Armstrong & Stillman, 2016*). The effects of combined stressor exposures range from synergistic to antagonistic when compared to single stressor exposure (*Rose, Warne & Lim, 2001*; *Fischer, Roffler & Eggen, 2012*; *Holmstrup et al., 2010*), and the scales and timing of response differ widely, rendering it challenging to predict the outcome of additional stressors (e.g. *Segner, Schmitt-Jansen & Sabater, 2014*).

However, such interactions between biotic stress and toxicants might be the rule rather than the exception and complicated indirect effects on predator-prey relationships seem to be common in aquatic communities (*Rohr & Crumrine, 2005*; *Langer-Jaesrich et al., 2010*; *Trekels, Van de Meutter & Stoks, 2013*).

Phenotypic differences within a population are another source of variation that complicates predicting multiple stressor effects. Recognizing this trait variation has profound and practical consequences for ecotoxicology, but also human medicine (*Evans & Relling, 2004*). Nevertheless, effects of stressors are often tested separately using laboratory populations of limited genetic diversity (*OECD, 2012*; *Macken, Lillicrap & Langford, 2015*). The use of unique strains and clones inherently misses the ecotoxicological target 'to predict effects in real populations' (*Lam & Gray, 2001*).

The tremendous number of possible stressor combinations and the influence of phenotypic variation on the biota response poses a grand challenge for ecotoxicology, which is seemingly impossible to deal with, and yet unavoidable. Gathering individual data is more laborious compared to gathering pooled or group data, especially when it comes to following the development of individuals. The sensitivity of individuals is likely highest at specific life stages and stage transitions. Therefore, our focus should be to identify these states rather than only addressing the larger time scale responses related to life span, fecundity, and adult survival. While the development rate of an animal can be estimated from daily observations, to actually measure growth parameters e.g. length or examine rapidly occurring ontogenetic events e.g. timing and duration of stage transitions, requires the frequent observer presence and potentially the repeated handling of the experimental animals. Even the mere observer presence may involuntarily alter the behaviour and development of individuals (*Mallet et al., 1987*; *Baker & McGuffin, 2007*). Hence using traditional methods to gather individual data, such as manually extracting, measuring, and placing them back into their holding container, can lead to biased results from greater effects of handling than treatment.

This challenge calls for the application of efficient and automated testing platforms to execute well-designed and manageable experiments that compare stress responses of organisms. In recent years, automated experimental systems have been developed to monitor water quality by quantifying motility and other endpoints in indicator species ranging from single-celled Euglenas (*Tahedl & Häder, 2001*; *Lee, Zheng & Yang, 2012*),

over Daphnia (*Häder & Erzinger, 2017*) to fish (*Cunha et al., 2008*). There is a range of automated systems which find use as in real-time monitoring of behavioural responses of aquatic organisms (reviewed in *Bae & Park, 2014*).

Among automated systems microfluidic Lab-on-a-chip systems (*Campana & Wlodkowic, 2018*) allow for the precise dosing in biochemical assays treatments. Due to their small size and cost efficiency they could be used in high-throughput screening of new chemicals. Measured endpoints range from bioluminescence production (*Zhao & Dong, 2013*), to viability (*Gammoudi et al., 2014*), and motility (*Huang, Campana & Wlodkowic, 2017*). One drawback is however that small volumes can limit the size of the testable animals (*Campana & Wlodkowic, 2018*) and might hinder a 'natural' response to the treatment, especially when measured throughout the complete development period of an organism. The aim of this study was thus to develop and test an affordable automated imaging system that allows for the continuous observation of a large number of separately kept individuals. We validated our approach by comparing our detailed data and results to the ones from *Lode et al. (2018)* which are based on temporally less resolved data from the same experiment.

We used copepods as model organisms, as they are key players in marine pelagic food webs and the most abundant metazoans on the planet (*Humes, 1994*; *Naganuma, 1996*). In recent years, copepods are also increasingly used as models in ecotoxicology (*Macken, Lillicrap & Langford, 2015*; *Raisuddin et al., 2007*).

The experiment followed the development of individual copepods under the influence of a biocide (copper) and a natural stressor in the form of chemical cues of a fish predator. In our case these include both kairomones, chemical cues emitted by the fish, which benefits the receiver and potentially harms the emitter, and Schreckstoff, cues from eaten copepods that can warn the other individuals. For simplicity we refer to these as predator cues in the remainder of the manuscript. We used a sublethal concentration of copper. We thus expected small but accumulative effects that would likely affect specific developmental stages. We anticipated that the highly resolved data would allow us to determine the most affected development stages, and also uncover subtle changes in growth trajectories and in the duration of stage transitions.

## MATERIALS AND METHODS

### Automated platform for image acquisition

We used an automatized imaging setup to follow individual growth at an hourly resolution, with a self-made experimental system that is capable of filming single culture plate wells repeatedly over the course of a copepod's development time. We used a DIY plotter kit (Makeblock Co., Ltd., Shenzhen, China: XY-plotter Robot Kit V2, see Fig. 1) as a basis for the system. On the movable platform, we mounted an upward facing infrared-capable camera with an image resolution of 2,592 × 1,944 pixels (Raspberry Pi NOIR with C mount and a six mm adjustable-focus lens). The images were saved as jpgs with a moderate compression of 85 to reduce file size while maintaining details, with 100 representing the maximum possible quality.

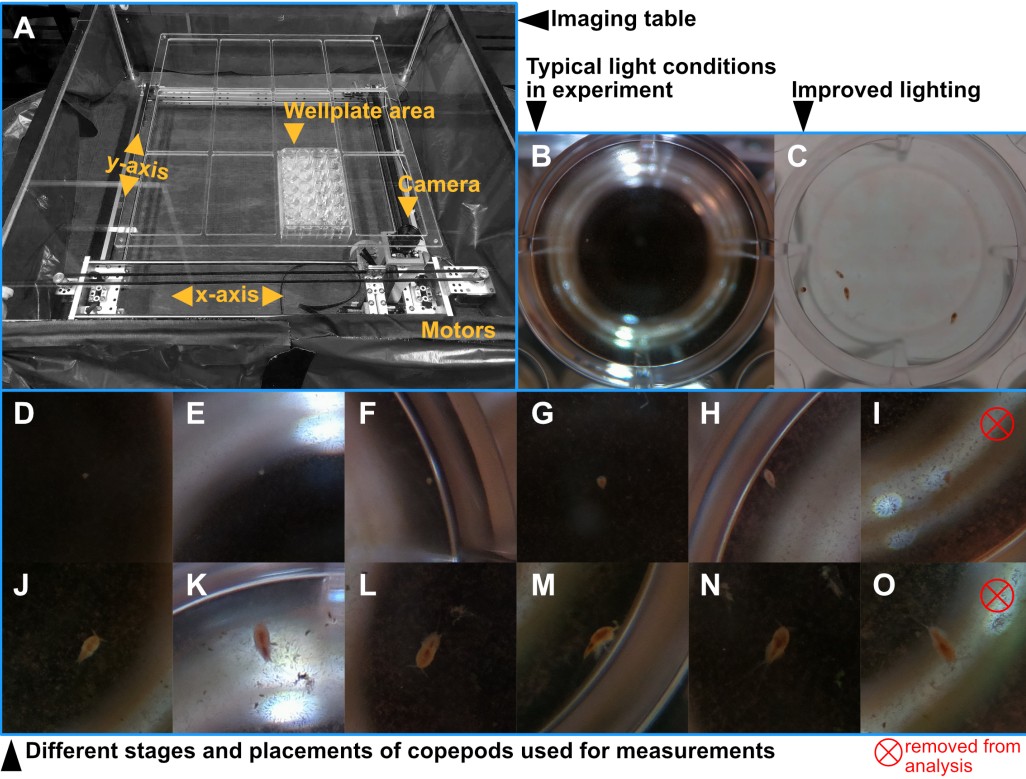

**Figure 1** **Visual description of the automated imaging setup and samples of the captured images.** (A) Photo of the automated imaging setup. The camera is placed on a movable platform below a transparent pane on which we placed well plates. The height of the pane can be adjusted to accommodate different sized well plates and magnifications. Two white LED light sources provided constant light from the sides of the table. (B) Example of the light conditions in the reported experiment, and (C) an example of improved light conditions when using an electroluminescent light sheet on top of the well plates. (D–H, J–N) Cropped images showing different copepod life stages that were used to measure the size of animals. Images in which the animal's outline was hard to distinguish from the background (I and O) would not be used for measuring due to the bad lighting condition and removed from the analysis.

Above the camera system, we installed a platform made of transparent Plexiglas. On this, we placed four 24-well plates containing the animals. Two stepper motors move the camera position. Instructions are taken from a python script on a microcomputer (Raspberry Pi), which also controls the camera. We programmed the system to sequentially take one image (2,592 × 1,944 pixels) of each well approximately every hour, for a period of 13 days. Two LED lamps (Camlink CL-Studio10) provided constant illumination to the setup from opposing sides, providing dark field imaging conditions. After each well plate the system reset its positioning system using two contact switches at each axis. This prevents a continuous systematic error in cases when the system gets misaligned. We later discovered that such errors occurred sporadically to be due to non-optimal baud rate settings of the serial port. The source code for the imaging system is included in Supplemental Material S1. The material costs of the system summed up to less than 500 EUROs.

The build of the setup is for the largest part easy as the plotter kit is targeted at juveniles. The execution of the script and adding changes to the number of plates,

timing of recordings can be done by any person with basic python knowledge. User-friendliness could however easily be improved by adding a simple graphical user interface to the script.

## Experiment and animals

We tested the platform with an experiment on combined effects of predator cues and copper exposure on copepod age and size at maturity.

We used the harpacticoid copepod *Tigriopus brevicornis* as the model organism. The laboratory stock cultures originated from a splash water pool in Drøbak (Norway) and one from Tjärnö (Sweden). We kept the stock cultures at 30 psu, 18 °C and a 12/12 light dark cycle for more than six months before the experiment. Stock cultures were fed ad libitum three times a week an equal mix of *Rhodomonas salina*, *Isocrysis galbana*, and *Dunaniella tertiolecta*.

Prior to the experiment, we picked single females with egg sacs from the stock culture. We placed them individually into 24 well plates. Every 30 min we checked manually if nauplii had hatched. If nauplii were detected, we registered the time, assigned a clutch ID, and removed the female. The nauplii were then placed individually in wells containing 2.5 ml of water with the respective treatment.

We exposed copepods to one of four treatments: predation risk (predator cues), copper (20 $\mu$g L$^{-1}$), combined predation risk and copper (20 $\mu$g L$^{-1}$), or control (pure seawater). All seawater was taken from the outer Oslofjord and filtered at 1.2 $\mu$m prior to use. We prepared seawater with predator cues by incubating three-spined stickleback (*Gasterosteus aculeatus*) for 48 h in filtered seawater at 18 °C, 30 psu, two fish l$^{-1}$. The fish were fed with *T. brevicornis* first at initiation, and once more after 24 h. Following incubation and removal of fish, the water was filtered (GF/C, 1.2 $\mu$m) and frozen (−18 °C). Water without predator cues was prepared similarly but without addition of fish and copepods. These frozen bottles were thawed daily to prepare the exposure solutions. For Cu and the combined treatment, we then added Cu through a 2-step dilution process of a 0.1 M CuSO$_4$ stock solution. Instead of Cu, we similarly added distilled water for the control and the predation risk treatment.

We replenished 72% of the exposure solution daily for each individual. We inspected the individuals daily using a binocular microscope and recorded survival, took a photo for subsequent length measurements, and most importantly assessed the development stage. We used the numbers of exuviae to determine the stage of the copepods. For a more detailed description of treatment preparation and the general procedure see *Lode et al. (2018)*.

## Image acquisition

We incubated 72 individual copepods from nine clutches of different mothers. One was lost during the setup of the experiment, and two could not be followed due to a misalignment of the robot. From the 69 other ones, we managed to extract on average 168.9 ± 31.1 measurements for each individual during development. The upper number of images and thus the maximum amount of measurements was 438.

## Length measurements

In total, we acquired 11,657 images that were suitable for measurements. Some of the other pictures were misaligned due to occasional glitches in the computer drives platform, or because the system took them during the daily water change, when the wells were removed from the platform (*Lode et al., 2018*) These images had to be removed from the analysis. We imported the images into the imaging software Fiji (*Schindelin et al., 2012*). The high magnification reduced the depth of field and the copepods could only be measured when they were in close proximity of the bottom (Fig. 1C). If the animal was distinctly visible and in focus, we manually measured its length. Due to the benthic lifestyle of the nauplii compared to the demersal behaviour of adults, we were able to obtain more measurements from younger stages. To compare the length measurement from machine images and the one obtained from microscope image by *Lode et al. (2018)*, we calculated a daily average of the machine length measurements.

## Moult from nauplii to copepodite

We determined the time of moult to the first copepodite stage (i.e. moment) by manually screening the pictures for the first occurrence of the copepodite stage. Earlier moults from nauplii to nauplii or moults from copepodite to copepodite were not as obvious and could not be easily determined directly from the images. Sometimes a water change, a temporary misalignment of the camera, or the temporary removal during the manual screening led to unusable pictures. If such unusable pictures preceded the first appearance of a copepodite stage, we were not able to accurately determine the time point. To be able to account for the varying degrees of uncertainty we noted down the number of 'uncertain' pictures, and included this information in the statistical analysis.

From the images we saw that most nauplii remained motionless for several frames during their transition to the first copepodite stage. To be able to determine whether the treatments influenced the moult duration to copepodite, we recorded the number of 'motion free' pictures before the first appearance of the copepodite. We included the first picture in which they were detected in the same position.

## Statistical analysis

All statistical analyses and data manipulation were conducted using the statistical software R (Version 3.5.0) (*R Core Team, 2018*). We used all acquired length measurements for the analyses. We converted the measurement time to the actual age of the individual using the time of birth (±30 min uncertainty). To meet normality assumptions, copepodite transition time and length data were log-transformed before analysis, and data of the duration of the nauplii to copepodite transition was square root transformed. If not otherwise stated, we used an information theoretic approach to select the best model for the linear models based on the corrected Akaike information criterion (AICc) for all measured response variables. If there were several models within a $\Delta$AICc < 2 of the best one, we averaged model estimates. We used the package MuMIn (*Bartoń, 2018*) for model selection and averaging.

To test whether length measurements from the images where comparable to the manual measurements from *Lode et al. (2018)*, we used a linear model with machine as dependent, microscope measurements as fixed factor, and individual ID as random factor to control for repeated measurements of the same individual. We also included treatment as fixed factor in the initial model, to test for differences in measurement 'accuracy' between treatments.

We tested the influence of treatment and clutch ID on transition timing from the last nauplii to the first copepodite stage with a two-way ANOVA. We allowed for an interaction between both independent factors in the initial model. We controlled for the uncertainty in transition time, by including the number of 'uncertain frames' as weights in the analysis.

To test whether males and females were affected differently by the treatments, we censored the data to include only matured individuals. We then examined the influence of treatment and gender in a separate model with these two factors as fixed factors, and copepodite transition time as the dependent variable. As before we allowed for all interactions between the independent factors and the number of uncertain frames as the weighting factor.

We tested for differences between clutch IDs and treatment in the moult duration of the nauplii to copepodite transition using a linear model with duration as dependent variable, and clutch ID and treatment as fixed factors. We allowed for interactions between the fixed factors in the initial model, and always included the number of uncertain frames as the weighting factor.

We tested for the influence of copper, predator cue, and clutch ID on length development using general additive models (GAM) with thin plate regression splines, using the mgcv-package (Version 1.8-23) for general additive modelling (*Wood, 2003, 2011*). Our models included individual ID as random factor, allowing for a random smoothing over time for each individual. In this case, to choose the best model, we started out with a model allowing for full interactions between the fixed factors and took advantage of the inbuilt model selection tool of the mgcv-package where the smoothing parameter estimation allows for model terms to approach zero. This procedure results in a final best fitting model. The final model parameters and smoothing functions were then evaluated using the function gam.check() and based on k`, estimated degrees of freedom (edf) and $p$-values. We further visually inspected the distribution of the residuals, quantile-quantile plots, and residual vs linear predictions.

Except for the moult from nauplii to copepodite, it was challenging to determine the exact time point from one nauplii stage to the next by manual (visual) inspection of the acquired images. Therefore, we tested for differences in transition timing between nauplii stages by analyzing the predicted individual growth increments over time. For this, we first ran a GAM that included random smoothers for each individual only. We then calculated isochronal body length predictions on a 72-minute resolution, similar to the one of the raw data. To not use the model beyond the data range, individual level length predictions were limited to the respective time the individuals spent in the experiment.

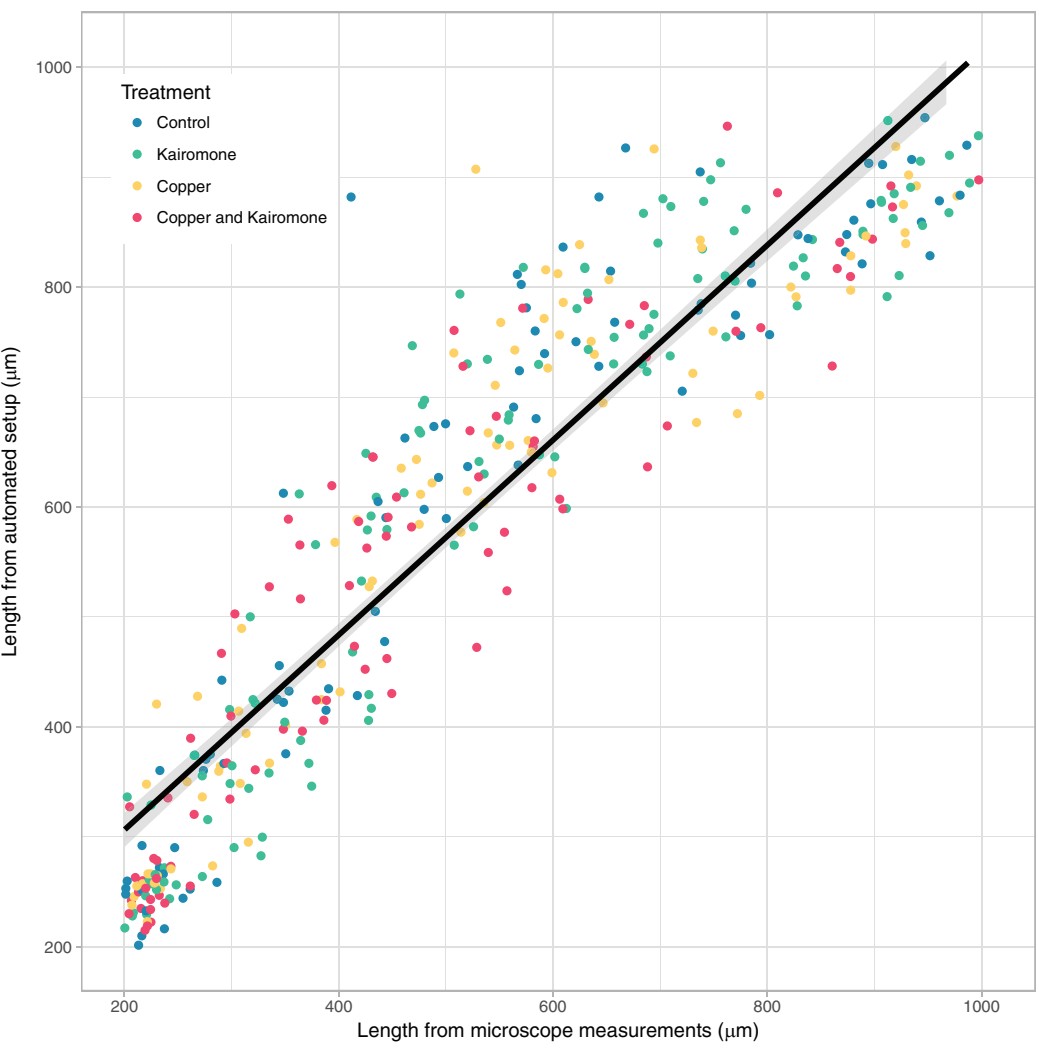

**Figure 2 Comparison of manual and automated measurements.** Comparison of copepod body lengths measured from machine images (daily average) and from daily microscope measurements acquired in *Lode et al. (2018)*. The line represents the prediction (± SE) from the linear regression.

From these, we derived the individual growth increments, which showed distinct peaks representing the stage transition phases.

We then used a new GAM with growth increments as the dependent factor to test for interactive effects of Clutch ID, copper, and predator cues on the transition dynamics. We also included negative predictions of growth to keep the normally distributed nature of the data and to emphasize the growth spurts during moults, although they are biologically impossible in this species.

To test whether copper, predator cues, or clutch ID influenced the final size of the copepods at the end of the experiment we restricted the data to measurements taken during the last day of the experiment and averaged them for each individual. We then analyzed this relationship using a linear model with copper, predator cues, and

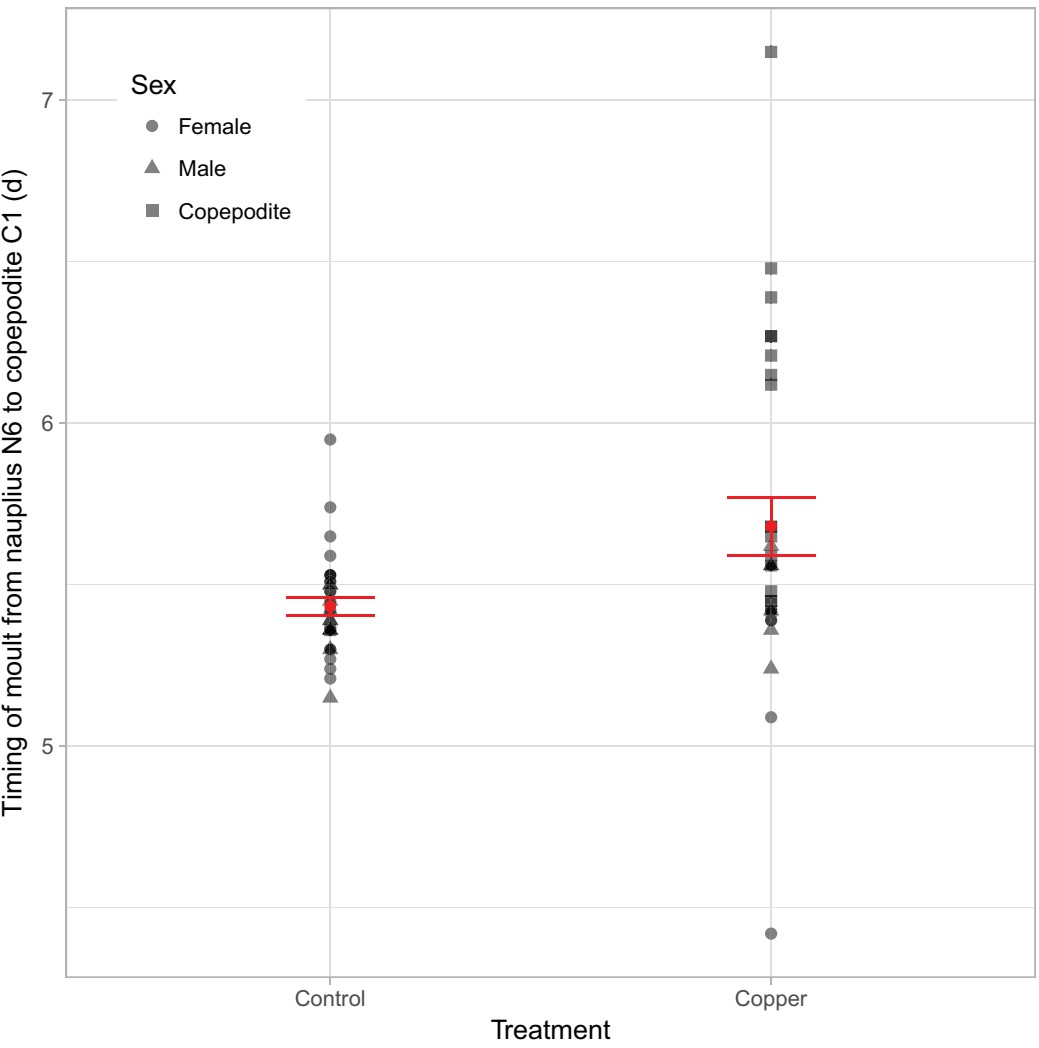

**Figure 3 Treatment effects on the timing of N6 to C1 metamorphosis.** The moment in time when nauplii N6 turned copepodites C1 depending on the copper treatment, with shapes indicating the sex as identified by the end of the experiment. Sex "Copepodites" are individuals that did not reach maturity by the end. The red dot and line indicate the mean and the SE of the raw data.

clutch ID as independent factors, and the log-transformed (averaged) body length as the dependent factor. In this case the endpoint size encompasses both potential effects of developmental delay and on size. To test whether treatment influenced only the final size of males and females differently we further restricted the averaged length data to include only lengths of fully matured individuals. We then used a similar model structure to the previous one but added gender as a fixed factor in the initial model.

## RESULTS

The daily manually measured length data by *Lode et al. (2018)* and data acquired using the robot table correlated well (estimate = 0.90, SE = 0.01, $F_{1,524}$ = 4,515, $p < 0.001$),

**Table 1 Model estimates of the best models testing the influence of copper, kairomone and clutch ID on the time of transition from nauplii N6 to copepodite C1 stage, as well as the duration of this major transition.**

| Data<br>Copepodite transition time | Factor | Estimate | Std. Error | | z-value | p-value |
|---|---|---|---|---|---|---|
| All individuals | (Intercept) | 1.69 | 0.03 | | 69.38 | <0.001 |
| | Copper | 0.14 | 0.04 | | 3.95 | 0.002 |
| **Duration of copepodite transition** | | **Estimate** | **Std. Error** | **Adj. SE** | **z-value** | **Pr(>\|z\|)** |
| All individuals | (Intercept) | 18.05 | 2.41 | 2.51 | 7.18 | <0.001 |
| | Copper | −2.20 | 1.24 | 1.29 | 1.70 | 0.089 |
| | Clutch D2 | 1.39 | 2.96 | 3.09 | 0.45 | 0.652 |
| | Clutch D3 | −4.96 | 2.92 | 3.04 | 1.63 | 0.103 |
| | Clutch S1 | −7.22 | 3.00 | 3.13 | 2.31 | 0.021 |
| | Clutch S2 | −8.35 | 3.03 | 3.16 | 2.64 | 0.008 |
| | Clutch S3 | −9.19 | 2.96 | 3.08 | 2.99 | 0.003 |
| | Clutch S4 | −4.42 | 2.92 | 3.04 | 1.46 | 0.146 |
| | Clutch S5 | −2.02 | 3.13 | 3.26 | 0.62 | 0.535 |
| | Clutch S6 | −1.69 | 2.89 | 3.01 | 0.56 | 0.574 |

**Note:**
Estimates for the duration of the copepodite transition are conditional averages from model averaging of the best competing models.

although automated length estimates were generally larger than the manual measurements, especially during the early copepodite stages (Fig. 2). Treatments did not influence the relationship between machine and manual measurements.

Copper delayed the time of the moult from nauplii N6 to copepodite C1, while the transition was independent of an individual's clutch ID (Fig. 3; Table 1). Model estimates for the N6 to C1 transition timing from nauplii to copepodite in males and females were similar, and the effect was driven by individuals that did not reach maturity by the end of the experiment (Fig. 3).

In contrast to the time of moult, the *moult duration* from nauplii N6 to copepodite C1 was mainly influenced by the individual's clutch ID in interaction with copper (Fig. 4; Table 1). Individuals of six clutches showed a reduced moult duration, while in the other clutches the duration was prolonged compared the control (Fig. 4). The duration of the moult ranges from the time between two subsequent recordings (~72 min) to more than 400 min.

The development of individuals, measured as individual growth, is influenced by a complex interactive effect of copper, predator cues, and clutch ID (Fig. 5; Fig. S1; Table 2). The treatments left two clutches unaffected (Fig. S1), while individuals in all other families responded with delayed growth. When we focus on the treatment effects, copper alone delayed development while predator cues did not have an impact. The combination of both led to a stronger delay in the late copepodite stages (Fig. 5).

The conversion of length measurements to growth increments showed distinct peaks which revealed the moults of the copepods, with variation in growth increments

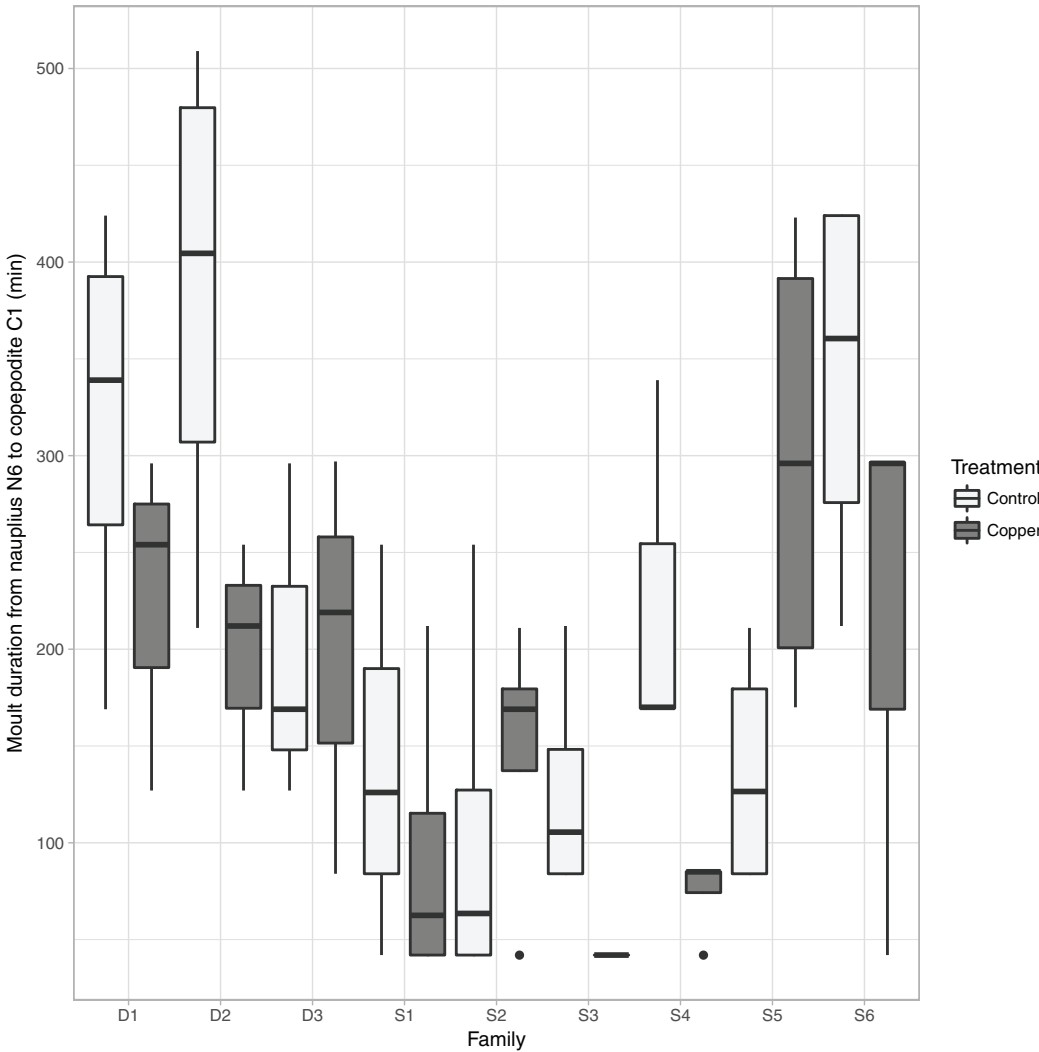

**Figure 4 Copper and clutch ID effects on moult duration.** Boxplot of the differences in N6 to C1 moult duration depending on clutch ID of the animals and the presence and absence of copper, the box shows median, quantiles, and the 1.5-time interquartile range is indicated by vertical lines.

explained by an interaction between Clutch ID, copper, and predator cues (Fig. 6; Table 3). Overall, the most significant delay in development occurs in individuals exposed to copper and predator cues combined when they metamorphose from nauplii to first copepodite (Fig. 6), which confirms the results of the manually screened timing of this major moult (Fig. 3; *Lode et al., 2018*). However already during the third transition the averaged peak height is reduced in the combined stressor treatment, which indicates a larger variability between exposed individuals. With an increased age of the individuals at the specific developmental stages, the moult cycles became less synchronized and measurement error became larger, and an overall trend between treatments was harder to detect, but also detecting individual peaks or moults becomes harder (Fig. 6). Therefore, we refrained from analyzing intermoult durations
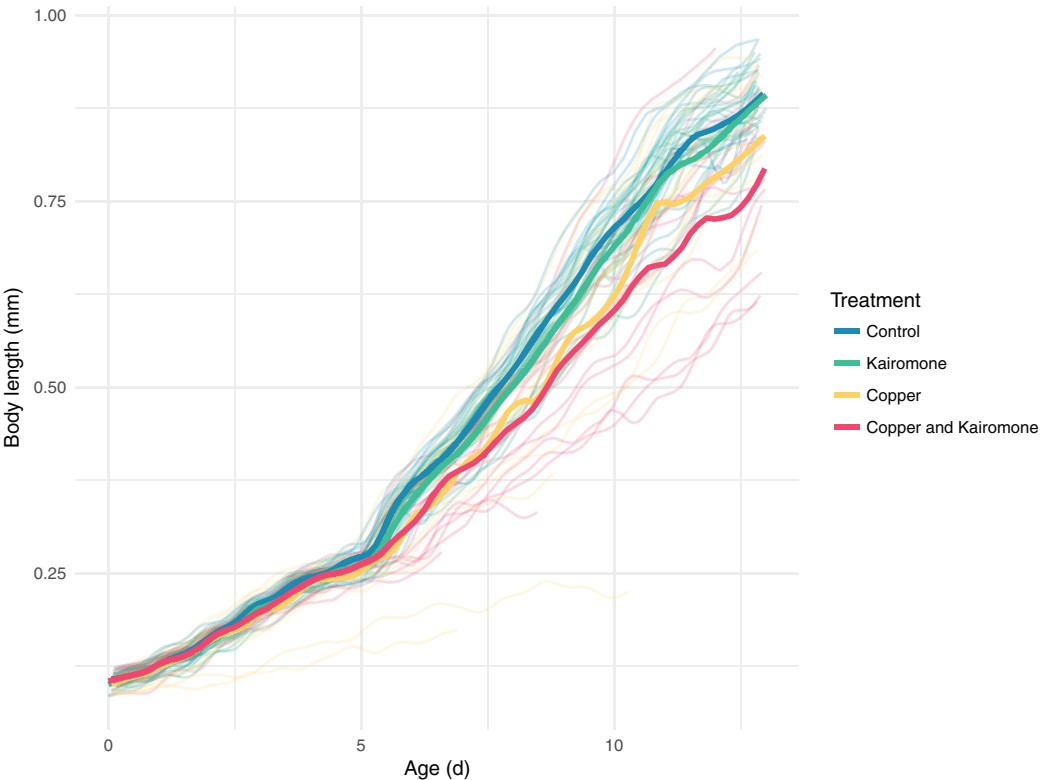

**Figure 5 Single and interactive effects of copper and kairomones on the development of *Tigriopus brevicornis*.** Faint lines indicate individual growth trajectories, while bold lines show the final GAM predictions averaged by treatment. While kairomones alone had little effect, the combination with copper reduced their growth in copepods more than did copper alone.

based on detected growth maxima. However, the peak heights of the averaged growth increments are highest in the control and predator cues treatment compared to the other two treatments, which means that there was less variation in transition timing in control and predator cues treatment. Or in other words, the effects of copper seem to be strongly affected by individual and clutch variation.

The final body size at the end of the experiment depends on additive effects of treatment and Clutch ID. Both competing models (ΔAICc < 2) included additive effects by copper and clutch ID, while only one had a negative effect of predator cues (Table 4). Some clutches were unaffected by treatment while for most others the exposure by both treatments led to reduced length (Fig. S1). However, the 23 unmatured copepodites at the end of the experiment in the copper and combined treatment biased these results. Therefore, we restricted the analyses to include mature individuals only ($n = 49$). The analysis showed several competing models (Table 4), which revealed that sex is the most important factor and to a lesser degree Clutch ID and copper. In general, males were slightly larger than females (Table 4).

**Table 2 Summary statistics of the final generalized additive model describing the influence of copper, kairomone and Clutch ID on growth over time.**

| Parametric coefficients | Estimate | Std. Error | *t* value | Pr(>|t|) |
|---|---|---|---|---|
| Intercept (Control Clutch ID D1) | −1.20 | 0.02 | −51.09 | <0.001 |
| Clutch ID D2 | 0.14 | 0.03 | 4.12 | <0.001 |
| Clutch ID D3 | 0.06 | 0.04 | 1.52 | 0.129 |
| Clutch ID S1 | 0.18 | 0.03 | 6.23 | <0.001 |
| Clutch ID S2 | 0.20 | 0.03 | 6.49 | <0.001 |
| Clutch ID S3 | 0.10 | 0.03 | 3.58 | <0.001 |
| Clutch ID S4 | 0.16 | 0.03 | 5.41 | <0.001 |
| Clutch ID S5 | 0.14 | 0.03 | 4.54 | <0.001 |
| Clutch ID S6 | 0.22 | 0.03 | 6.41 | <0.001 |
| Kairomone | −0.01 | 0.02 | −0.58 | 0.562 |
| Copper | −0.11 | 0.02 | −6.58 | <0.001 |

| Approximate significance of smooth terms Factor combination | edf | Ref.df | *F*-value | *p*-value |
|---|---|---|---|---|
| No Copper/No Kairomone/Clutch ID D1 | 1.95 | 39.00 | 2.23 | <0.001 |
| Copper/No Kairomone/Clutch ID D1 | 0.99 | 18.00 | 4.64 | <0.001 |
| No Copper/Kairomone/Clutch ID D1 | 0.99 | 14.00 | 5.97 | <0.001 |
| Copper/Kairomone/Clutch ID D1 | 3.15 | 39.00 | 1.97 | <0.001 |
| No Copper/No Kairomone/Clutch ID D2 | 5.55 | 39.00 | 2.39 | <0.001 |
| Copper/No Kairomone/Clutch ID D2 | 3.38 | 39.00 | 2.32 | <0.001 |
| No Copper/Kairomone/Clutch ID D2 | 3.88 | 39.00 | 3.06 | <0.001 |
| Copper/Kairomone/Clutch ID D2 | 0.99 | 14.00 | 5.83 | <0.001 |
| No Copper/No Kairomone/Clutch ID D3 | 2.93 | 39.00 | 2.58 | <0.001 |
| Copper/No Kairomone/Clutch ID D3 | 4.74 | 35.00 | 2.64 | <0.001 |
| No Copper/Kairomone/Clutch ID D3 | 2.62 | 39.00 | 2.57 | <0.001 |
| Copper/Kairomone/Clutch ID D3 | 3.61 | 39.00 | 2.37 | <0.001 |
| No Copper/No Kairomone/Clutch ID S1 | 1.94 | 39.00 | 1.03 | <0.001 |
| Copper/No Kairomone/Clutch ID S1 | 0.99 | 18.00 | 4.26 | <0.001 |
| No Copper/Kairomone/Clutch ID S1 | 2.44 | 39.00 | 2.25 | <0.001 |
| Copper/Kairomone/Clutch ID S1 | 0.98 | 12.00 | 5.43 | <0.001 |
| No Copper/No Kairomone/Clutch ID S2 | 2.92 | 39.00 | 2.34 | <0.001 |
| Copper/No Kairomone/Clutch ID S2 | 2.83 | 39.00 | 2.12 | <0.001 |
| No Copper/Kairomone/Clutch ID S2 | 3.51 | 39.00 | 2.57 | <0.001 |
| Copper/Kairomone/Clutch ID S2 | 0.99 | 10.00 | 7.16 | <0.001 |
| No Copper/No Kairomone/Clutch ID S3 | 0.99 | 13.00 | 5.80 | <2e-16 |
| Copper/No Kairomone/Clutch ID S3 | 0.99 | 14.00 | 5.73 | <2e-16 |
| No Copper/Kairomone/Clutch ID S3 | 1.68 | 39.00 | 2.04 | <2e-16 |
| Copper/Kairomone/Clutch ID S3 | 2.18 | 39.00 | 1.46 | <2e-16 |
| No Copper/No Kairomone/Clutch ID S4 | 2.43 | 39.00 | 1.10 | <2e-16 |
| Copper/No Kairomone/Clutch ID S4 | 0.99 | 16.00 | 4.59 | <2e-16 |
| No Copper/Kairomone/Clutch ID S4 | 2.84 | 39.00 | 2.26 | <2e-16 |

(*Continued*)

**Approximate significance of smooth terms**

| Factor combination | edf | Ref.df | F-value | p-value |
|---|---|---|---|---|
| Copper/Kairomone/Clutch ID S4 | 0.99 | 10.00 | 7.63 | <2e-16 |
| No Copper/No Kairomone/Clutch ID S5 | 3.15 | 39.00 | 2.71 | <2e-16 |
| Copper/No Kairomone/Clutch ID S5 | 0.99 | 12.00 | 7.31 | <2e-16 |
| No Copper/Kairomone/Clutch ID S5 | 3.20 | 39.00 | 2.65 | <2e-16 |
| Copper/Kairomone/Clutch ID S5 | 2.27 | 39.00 | 2.00 | <2e-16 |
| No Copper/No Kairomone/Clutch ID S6 | 3.80 | 39.00 | 2.77 | <2e-16 |
| Copper/No Kairomone/Clutch ID S6 | 2.32 | 39.00 | 2.28 | <2e-16 |
| No Copper/Kairomone/Clutch ID S6 | 3.13 | 39.00 | 2.55 | <2e-16 |
| Copper/Kairomone/Clutch ID S6 | 0.97 | 8.00 | 4.78 | <2e-16 |
| Well | 1312.40 | 2677.00 | 8.78 | <2e-16 |

**Note:**
The model was formulated as Length ~ Clutch ID + Kairomone + Copper + te(age, by = interaction(Copper, Kairomone, Clutch ID), $k = 40$) + s(age, Well, bs = "fs", $m = 1$, $k = 40$). In the table edf represents the number of effective degrees of freedom and Ref.df the reference number of degrees of freedom used for hypothesis testing.

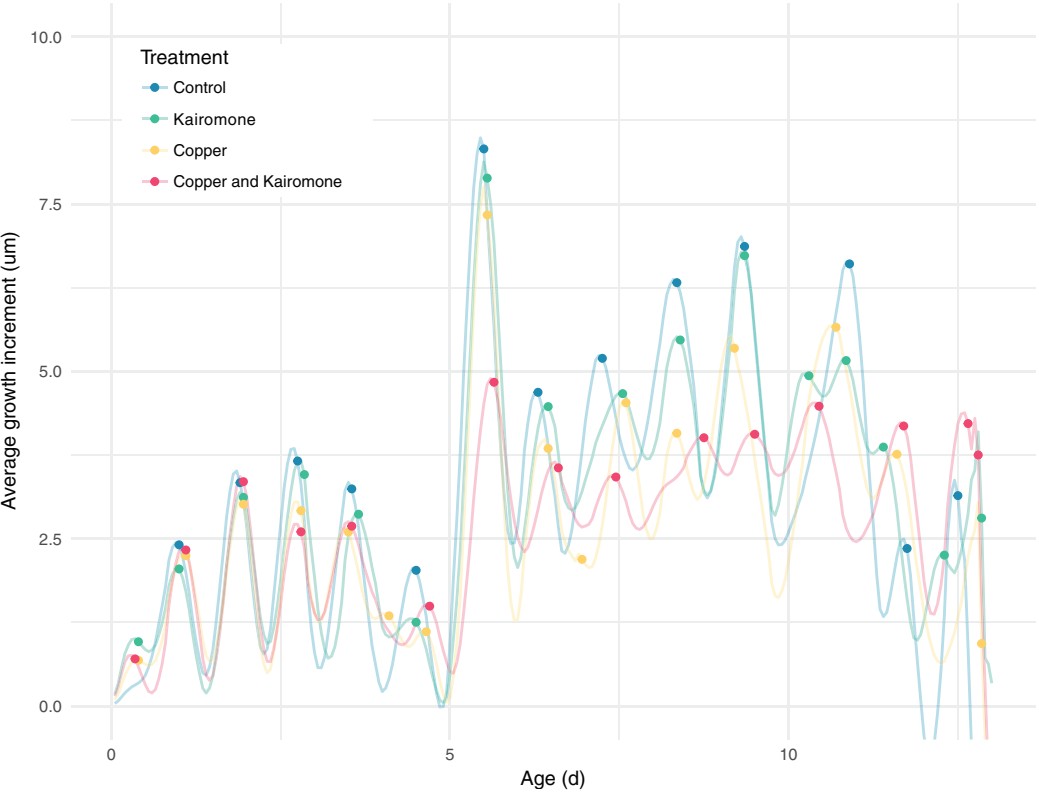

**Figure 6 Predicted individual growth increments as a function of treatment combination.** The individual predictions are averaged over each treatment combination and time point with a 72-min resolution. A lower average growth increment indicates a larger variability in transition timing between individuals.

**Table 3 Summary table for of the best general additive model describing the growth dynamics of individual copepods over age.**

| Parametric coefficients | Estimate | Std. Error | t-value | Pr(>|t|) |
|---|---|---|---|---|
| (Intercept) | 0.0035 | 0.0051 | 0.70 | 0.483 |
| Kairomone | 0.0043 | 0.0027 | 1.63 | 0.102 |
| Clutch IDD2 | −0.0011 | 0.0057 | −0.19 | 0.851 |
| Clutch IDD3 | −0.0056 | 0.0065 | −0.87 | 0.395 |
| Clutch IDS1 | −0.0031 | 0.0066 | −0.48 | 0.634 |
| Clutch IDS2 | −0.0072 | 0.0057 | −1.26 | 0.209 |
| Clutch IDS3 | −0.0013 | 0.0057 | −0.24 | 0.814 |
| Clutch IDS4 | −0.0067 | 0.0066 | −1.01 | 0.312 |
| Clutch IDS5 | −0.0041 | 0.0054 | −0.77 | 0.444 |
| Clutch IDS6 | 0.00074 | 0.0061 | 0.12 | 0.904 |
| Copper | 0.00065 | 0.0028 | 0.23 | 0.816 |

| Approximate significance of smooth terms | edf | Ref.df | F | p-value |
|---|---|---|---|---|
| No Kairomone/Clutch ID D1/No Copper | 30.63 | 39.00 | 9.05 | <0.001 |
| Kairomone/Clutch ID D1/NoCopper | 16.78 | 39.00 | 1.27 | <0.001 |
| No Kairomone/Clutch ID D2/No Copper | 27.85 | 38.00 | 7.21 | <0.001 |
| Kairomone/Clutch ID D2/No Copper | 32.20 | 39.00 | 15.70 | <0.001 |
| No Kairomone/Clutch ID D3/No Copper | 35.13 | 39.00 | 40.36 | <0.001 |
| Kairomone/Clutch ID D3/No Copper | 26.48 | 39.00 | 8.05 | <0.001 |
| No Kairomone/Clutch ID S1/No Copper | 23.27 | 39.00 | 4.88 | <0.001 |
| Kairomone/Clutch ID S1/No Copper | 13.07 | 38.00 | 1.75 | <0.001 |
| No Kairomone/Clutch ID S2/No Copper | 19.24 | 39.00 | 1.83 | <0.001 |
| Kairomone/Clutch ID S2/No Copper | 0.14 | 39.00 | 0.00 | 0.107 |
| No Kairomone/Clutch ID S3/No Copper | 28.26 | 38.00 | 11.56 | <0.001 |
| Kairomone/Clutch ID S3/No Copper | 0.00 | 39.00 | 0.00 | 0.366 |
| No Kairomone/Clutch ID S4/No Copper | 29.75 | 38.00 | 20.54 | <0.001 |
| Kairomone/Clutch ID S4/No Copper | 18.77 | 38.00 | 4.48 | <0.001 |
| No Kairomone/Clutch ID S5/No Copper | 0.00 | 39.00 | 0.00 | 0.189 |
| Kairomone/Clutch ID S5/No Copper | 0.00 | 39.00 | 0.00 | 0.199 |
| No Kairomone/Clutch ID S6/No Copper | 11.35 | 39.00 | 0.91 | <0.001 |
| Kairomone/Clutch ID S6/No Copper | 28.32 | 39.00 | 7.90 | <0.001 |
| No Kairomone/Clutch ID D1/Copper | 25.90 | 39.00 | 5.83 | <0.001 |
| Kairomone/Clutch ID D1/Copper | 8.09 | 38.00 | 0.60 | <0.001 |
| No Kairomone/Clutch ID D2/Copper | 0.00 | 39.00 | 0.00 | 0.128 |
| Kairomone/Clutch ID D2/Copper | 27.19 | 39.00 | 8.08 | <0.001 |
| No Kairomone/Clutch ID D3/Copper | 0.00 | 35.00 | 0.00 | 0.540 |
| Kairomone/Clutch ID D3/Copper | 25.68 | 39.00 | 8.11 | <0.001 |
| No Kairomone/Clutch ID S1/Copper | 17.90 | 38.00 | 1.95 | <0.001 |
| Kairomone/Clutch ID S1/Copper | 20.44 | 39.00 | 4.03 | <0.001 |
| No Kairomone/Clutch ID S2/Copper | 11.04 | 39.00 | 0.95 | <0.001 |
| Kairomone/Clutch ID S2/Copper | 27.59 | 39.00 | 4.62 | <0.001 |

(Continued)

**Table 3** (*continued*).

| Approximate significance of smooth terms | edf | Ref.df | F | *p*-value |
|---|---|---|---|---|
| No Kairomone/Clutch ID S3/Copper | 33.77 | 39.00 | 35.25 | <0.001 |
| Kairomone/Clutch ID S3/Copper | 23.20 | 39.00 | 4.64 | <0.001 |
| No Kairomone/Clutch ID S4/Copper | 24.30 | 39.00 | 6.16 | <0.001 |
| Kairomone/Clutch ID S4/Copper | 29.63 | 39.00 | 15.37 | <0.001 |
| No Kairomone/Clutch ID S5/Copper | 17.67 | 38.00 | 1.59 | <0.001 |
| Kairomone/Clutch ID S5/Copper | 24.99 | 39.00 | 9.31 | <0.001 |
| No Kairomone/Clutch ID S6/Copper | 1.21 | 39.00 | 0.05 | 0.029 |
| Kairomone/Clutch ID S6/Copper | 11.60 | 39.00 | 0.71 | <0.001 |
| Well | 2244.00 | 2706.00 | 322.04 | <0.001 |

**Note:**
The model formulation was: Growth increment ~ Kairomone + Clutch ID + Copper + te(age, by = interaction (Kairomone, Clutch ID, Copper), $k = 40$) + s(age, Well, bs = "fs", $m = 1$, $k = 40$). In the table edf represents the number of effective degrees of freedom and Ref.df the reference number of degrees of freedom used for hypothesis testing.

**Table 4 Estimates for the final size (log-transformed data) of the copepods at the end of the experiment.**

| Final body length | Factor | Estimate | Std. Error | *z*-value | *p*-value |
|---|---|---|---|---|---|
| All individuals | (Intercept) Clutch D1 | −0.24 | 0.04 | 5.46 | <0.001 |
| | Copper | −0.10 | 0.02 | 4.03 | <0.001 |
| | Clutch D2 | 0.06 | 0.05 | 1.03 | 0.305 |
| | Clutch D3 | 0.08 | 0.06 | 1.25 | 0.211 |
| | Clutch S1 | 0.14 | 0.05 | 2.53 | 0.011 |
| | Clutch S2 | 0.11 | 0.05 | 2.23 | 0.027 |
| | Clutch S3 | 0.02 | 0.05 | 0.42 | 0.671 |
| | Clutch S4 | 0.15 | 0.05 | 0.05 | 0.007 |
| | Clutch S5 | 0.13 | 0.05 | 0.05 | 0.010 |
| | Clutch S6 | 0.18 | 0.05 | 0.05 | <0.001 |
| | Kairomone | −0.03 | 0.02 | 0.02 | 0.208 |
| Matured individuals | (Intercept) Clutch D1 | −0.18 | 0.03 | 5.32 | <0.001 |
| | SexM | 0.04 | 0.02 | 2.13 | 0.033 |
| | Clutch D2 | 0.05 | 0.03 | 1.60 | 0.109 |
| | Clutch D3 | 0.02 | 0.04 | 0.62 | 0.539 |
| | Clutch S1 | 0.10 | 0.03 | 2.93 | 0.003 |
| | Clutch S2 | 0.07 | 0.03 | 2.21 | 0.027 |
| | Clutch S3 | 0.04 | 0.04 | 1.08 | 0.280 |
| | Clutch S4 | 0.08 | 0.03 | 2.48 | 0.013 |
| | Clutch S5 | 0.10 | 0.03 | 3.01 | 0.003 |
| | Clutch S6 | 0.10 | 0.03 | 3.02 | 0.003 |
| | Copper | 0.01 | 0.02 | 0.82 | 0.412 |

**Note:**
Estimates are conditional model averages of the best competing models.

## DISCUSSION

The intensity of adverse responses to toxicants in the natural environment is challenging to predict due to the almost infinite number of possible interactions with biotic and abiotic factors (*Segner, Schmitt-Jansen & Sabater, 2014*), and begs for efficient methods to handle many replicates. In this study, we used an automated imaging approach to measure the combined impact of a toxicant (copper) and a biotic stressor (predator cues) on copepod development. We validated and evaluated the added benefits of our approach with the findings obtained using traditional manual methods at a much lower time resolution (*Lode et al., 2018*). We find the same complex interactions between the copper and predator cues treatment, and individual's clutch ID, in determining the growth trajectory of an individual facing multiple stressors.

Compared to the daily measurements of *Lode et al. (2018)* our highly resolved data allowed us to zoom in on individual moult events, which is not possible for large sample sizes using traditional methods. Especially during the naupliar stages, we detected clear peaks of moult events. These revealed that the treatment effects first affected the N3 transition and got more pronounced from then onwards. The biggest effect is visible during the naupliar to copepodite (N6-C1) metamorphosis. The significant differences between individuals and different clutches in their response to the treatments led to a wider distribution of the moult timings (Fig. 6). The strong influence of an individual's clutch ID on the major intermoult duration also suggests a genetic role in the resistance to multiple stressors, which is a major challenge in ecotoxicology (*Evenden & Depledge, 1997*; *Wirgin & Waldman, 2004*).

The concurrent results show the potential of our semi-automated system to tackle large sample sizes and detect small developmental differences in individual organisms, while still reducing the workload and the handling of the animals. Our setup can thus ease the collection of individual trait data and be used to answer questions in both toxicology and ecology. A focus on trait-based responses is especially helpful in studying the responses to multiple stressors. While we used it for small aquatic invertebrates, the imaging system is customizable and adjustable to accommodate different container- and species sizes. It thus is in line with the successful use of automatic monitoring systems in ecotoxicology like for example the Multispecies Freshwater BiomonitorTM (*Gerhardt & Schmidt, 2002*), LeDaphNet (*Lagergren et al., 2017*), and DaphniaTox (*Häder & Erzinger, 2017*) and lab-on-a-chip systems (*Zhu et al., 2015*; *Campana & Wlodkowic, 2018*).

In its current state, the system can reliably capture the size and movement of animals with a primarily benthic lifestyle. Examples include surface cruising animals such as benthic copepods, snails, trematodes, and nematodes. Using a bottom mounted camera works best when individuals are close to the bottom of the holding container, or in our case the well plate. Potential research questions include testing the influence of abiotic and biotic factors on the settlement of planktonic larvae of benthic animals, growth development of invertebrates at different nutrient concentrations, egg hatching times, and in this context also the factors which drive the emergence of resting eggs. Our system

can even quantify the behavioural variability in populations from recorded movie sequences. As individual variability is the foundation and currency of personality research, i.e. the study focusing on repeatable and correlated behaviours, the imaging system can easily be used to capture consistent differences between individuals and in consequence 'personalities' of animals. In recent years it has become clear that such characteristics that are not only present in 'higher' organisms but also in invertebrates (*Kralj-Fišer & Schuett, 2014*; *Sih, Bell & Johnson, 2004*).

The depth of field of the camera is one of the apparent drawbacks of the current system. With increasing age, the copepods became increasingly active and explored the whole water column. Thus, the number of images from which we could reliably determine body length decreased with age. This problem could be solved by using a smaller aperture or by adding a servomotor to the camera. Moving the plane of focus while recording a movie, would increase the chances of acquiring a sharp and complete image of the animal.

While the recording of the animals is automatized, our approach currently still relies on the manual screening and measuring of the images. This step is necessary due to the imperfect image quality (variable light conditions, occasional blurriness). An even background illumination using electroluminescent sheets or diffuse LED light sources does however increase the image quality to a point where it would be possible to implement image analyses based on neural networks. For example, using the tensorflow library (*Abadi et al., 2016*) we could then automatically classify stage data, measure size data, and other traits (gut content). The type of images could also be analyzed using crowd-based annotation services such as Quanti.us (*Hughes et al., 2018*).

## CONCLUSIONS

Our results illustrate the need to study the interactive effects of natural and anthropogenic stressors, and they underscore the necessity to consider the phenotypic and genetic variation in stress response if we want to use ecotoxicological studies to predict the consequences of toxicants for natural populations. Our system takes the idea of autosamplers, lab-on-a chip, and other high-throughput ideas, and applies it to questions related to the development and potentially the behaviour of small invertebrates. It uncovered differences in moult duration and the timing of copepod metamorphosis which would be difficult to detect using manual approaches. Given that it is easy to build, affordable, and runs with open source imaging and analysis software, it can be scaled to accommodate for high-throughput testing of multiple treatment combinations and gradients. When data are gained at both individual and population levels, they can be combined conceptually in adverse outcome pathways and increase the value of risk assessment in ecotoxicology (*Kramer et al., 2011*).

## ACKNOWLEDGEMENTS

We thank Rita Amundsen for algae culture maintenance, the workshop at IBV for help with building the experimental setup, and the UiO Aquaria facilities, in particular Haaken Hveding Christensen, for keeping the sticklebacks.

### Funding

This is a contribution to the prioritized research group LUMS (Life history under multiple stressors), financed by the Department of Biosciences, UIO. The funders had no role in study design, data collection and analysis, decision to publish, or preparation of the manuscript.

### Grant Disclosures

The following grant information was disclosed by the authors:
This is a contribution to the prioritized research group LUMS (Life history under multiple stressors).
Financed by the Department of Biosciences, UIO.

### Competing Interests

The authors declare that they have no competing interests.

### Author Contributions

- Jan Heuschele conceived and designed the experiments, performed the experiments, analyzed the data, prepared figures and/or tables, authored or reviewed drafts of the paper, approved the final draft.
- Torben Lode conceived and designed the experiments, performed the experiments, analyzed the data, authored or reviewed drafts of the paper, approved the final draft.
- Tom Andersen conceived and designed the experiments, analyzed the data, contributed reagents/materials/analysis tools, authored or reviewed drafts of the paper, approved the final draft.
- Katrine Borgå conceived and designed the experiments, contributed reagents/materials/analysis tools, authored or reviewed drafts of the paper, approved the final draft.
- Josefin Titelman conceived and designed the experiments, contributed reagents/materials/analysis tools, authored or reviewed drafts of the paper, approved the final draft.

### Data Availability

The raw length measurements are available in Supplementary Material S2.

### Supplemental Information

Supplemental information for this article can be found online at http://dx.doi.org/10.7717/peerj.6776#supplemental-information.

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
