# Peer review of "An affordable and automated imaging approach to acquire highly resolved individual data—an example of copepod growth in response to multiple stressors"

_PeerJ, doi:10.7717/peerj.6776_

## Round 0.1 · original submission · Major Revisions

Although one of the reviewers (#2) has recommended that the manuscript be rejected, I believe those comments/concerns can be addressed by providing additional details on the automated approach (as indicated by the other 2 reviewers) as well as some additional references. Please address all the reviewer comments in your revision; the reviews are very thoughtful overall, and will improve the manuscript.

·

Basic reporting

No comment

Experimental design

No comment

Validity of the findings

No comment

Additional comments

An affordable and automated imaging approach to acquire highly-resolved individual data - an example of copepod growth in response to multiple stressors (#33090)

The authors examined the combined effects of fish-released kairomones and copper toxicity on copepod life traits. The unique method of the automated imaging platform used for this analysis enables to view and analyze rapidly occurring changed in multiple individuals while reducing the bias originating from human interruption. The system has great potential to explain life history shifts of various species and as such would be of interest to ecologists from several fields.
I think the manuscript would benefit greatly from a figure summarizing the combine effects on copepod life-history traits. Currently, the effects are complex and not easy to follow.
Line 53: et al.
Lines 75-78: this sentence needs to be rephrased. I understand the point that the authors are trying to make, however constant observation is not required. I’ve conducted several such studies using mosquito larvae, and development rate can be calculated with 2 observations per day.
Line 95: The aim?
Line 96: sentence not clear, you mention that the system can follow a large number of individuals but do not mention that they need to be in separate wells.
Line 103: Define kairomone.
Line 137: Please expand on the kairomone treatment. What fish species? From what volume were the water taken, how many fish were placed there etc. I also understand that you used salt water. Please expand on how this was done (sea water or freshwater with additional salt?).
Line 281: AICc?
Line 287: Sex, not gender.
Line 299: you keep mixing the terms “kairomones” and “predator cues”, be constant.

Reviewer 2 ·

Basic reporting

Measurements for copepod growth rate are labor intense and time consuming. The authors developed an automated imaging platform to acquire temporally highly resolved individual data and tested this platform by exposing copepods to a combination of a biotic stressor (kairomone, i.e. predator cues) and a toxicant (copper) and measured the growth response of individual copepod. The content for this manuscript is heavy and the timeframe does not allow an in-depth review. My general impression is that this manuscript is likely to get published eventually, but the current format is not ready for publish.

When I read the title for the manuscript, I thought this would be a manuscript on a new methodology or new instrument. But when I read the manuscript, it appeared that this manuscript focused on their experimental results on the impact of a biotic stressor and toxicant on copepod growth. It is not a good choice to do two things in one attempt.

Experimental design

This “new” method and instrument was briefly mentioned in the manuscript. The authors did not give enough details and I found it was difficult to understand how the instrument works.

1) Is light source operated in continuous mode or pulse mode?
2) Is the depth field adequate for the well? If not, does it mean you have to manually acquire proper image for each copepod? That could very time consuming.
3) What is the pixel resolution for the camera? Does it provide enough resolution for small nauplii?
4) Please provide an acquired image with nauplii with proper legend.

Validity of the findings

The validation of new instrument is problematic. The authors compared their measurements against the “published data”. It does not look like the published was from not the same experiment and same individuals.

1) In figure 2, in which the authors intended to present their results against published results as they stated in figure legend. But the published results are nowhere to be found.

Additional comments

I apologize for the authors that I could not deliver a good news for their efforts. It does require extraordinary efforts to develop a new method and validate it against existing methods. This manuscript clearly falls short on both ends.

Two minor comments:

The numbers in the table and text should be consistent in terms of decimal point. Abbreviations should be spelled out in the table caption.

For the experiment design, it looks like that the authors performed their experiments on different clutches, but all data were pulled together to run their GAMs. This operation is likely valid when they want to examine clutch effect but may not valid when they examined overall chemical effects. It should be proceeded more carefully.

Reviewer 3 ·

Basic reporting

Clear and unambiguous, professional English used throughout.

Additional literature references required

Some additional Figures would be recommended

See "General comments for the author" for detailed remarks

Experimental design

Research question well defined, relevant & meaningful.

Some expansion of the results section is required

See "General comments for the author" for detailed remarks

Validity of the findings

Conclusion are well stated, linked to original research question & limited to supporting results

Data is robust, statistically sound, & controlled.

See "General comments for the author" for detailed remarks

Additional comments

The manuscript by Heuschele et al describes development of a novel, temporally resolved phenotypic analysis platform for the analysis of copepod growth in ecotoxicological bioassays.

I agree with the authors that the automated analysis in ecotoxicology is still underdeveloped and in fact ecotoxicology is one of the least automated fields of biosciences at this moment. More importantly the analysis of temporal events in environmental toxicology is rarely taken into account that is highly surprising since the physiological effects of contaminants depend not only on their concentration but also exposure time.

The foundations of this study are thus sound and there is definitely a need for more automated imaging (and non-imaging) high-throughput phenotypical analysis platforms to acquire highly resolved temporal data in ecotoxicology. The study is in general well written with sufficient methodological, experimental and analytical rigor. The data sets are clearly laid out and discussed well.

Areas for improvement are as follows:

1) Introduction – is a bit complex and in part a bit off the main topic (especially in the first part).
Authors should focus on:
i) importance of the phenotypical analysis, high-throughput phenomics (as a future in ecotoxicology in general)
ii) review of the most recent technological advanced in automation of ecotoxicological assays
iii) finally position their system accordingly to the above and outline innovation relating to copepod bioassays performed at the moment

2) A quick search of the web indicates some recent papers that should be cited/discussed and that in fact pioneered the concepts of automated phenotypical analysis, time resolved analysis, and even miniaturization in aquatic ecotoxicology:
- O Campana, D Wlodkowic (2018) Environmental science & technology 52 (3), 932-946
- O Campana, D Wlodkowic (2018) Sensors and Actuators B: Chemical 257, 692-704
- Y Huang, O Campana, D Wlodkowic (2017) Scientific reports 7 (1), 17603
- F Zhu, A Wigh, T Friedrich, A Devaux, S Bony, D Nugegoda, J Kaslin, D Wlodkowic (2015) Environmental science & technology 49 (24), 14570-14578
- J Akagi, F Zhu, CJ Hall, KE Crosier, PS Crosier, D Wlodkowic (2014) Cytometry Part A 85 (6), 537-547

3) Materials and Methods – camera system should be described in more details including resolution, type of files generated compression, codecs etc.

4) Results – The description of the system design, validation and operating principles should be provided in details at the beginning of the results section since this is the main innovation of this work.

5) Appropriate diagrams and flow charts of the operation of the system should be provided together with photographs of the prototype. The Figure 1 is too minimalistic!

6) As per the above (see comment 3) the authors should describe and possible present numerical data to demonstrate the reliability and reproducibility of the XY-plotter to position the camera repeatedly over the same wells over time. How did the authors assessor that there was no drift from X-Y coordinates. Were additional sensors employed, was there an optical algorithm detecting an outline of the well employed?

7) Exemplary micrographs generated by the system should be also demonstarted in one of the Figures.

8) Why on Figure 1 light sources are positioned horizontally on both ends of the system. From my experience a light box with a carefully designed diffuser etc is usually needed to provide a uniform illumination of samples. Authors should discuss this in much more details. Did they use any system to distribute the light evenly over all samples? How did authors minimize the edge shadow effect due to a meniscus of media in the wells?

9) Discussion “Our system can even quantify the behavioral variability in 

populations from recorded movie sequences. “ – unlikely since the time delay the current system requires to move camera from well 1 to well x will introduce temporal gaps in data. For behavioral analysis single ultra resolution camera system is required to capture movies simultaneously from all samples.

10) Authors could also briefly describe costs of the construction of a similar system and its user friendliness for laboratories without any engineering inclination.

---

## Round 0.2 · accepted · Accept

Thank you for your efforts in revising your manuscript.

# Reviewer 3 ·

Basic reporting

I have no further comments. Authors have addressed my previous comments in this revised version.

Experimental design

I have no further comments. Authors have addressed my previous comments in this revised version.

Validity of the findings

I have no further comments. Authors have addressed my previous comments in this revised version.

Additional comments

I have no further comments. Authors have addressed my previous comments in this revised version.